# Perturbation Analysis of Travel-Time Accuracy for Core Phases Reconstructed from Seismic Interferometry

Yingjie Xia<sup>1,2,3</sup>, Xuping Feng<sup>2,3</sup>, and Xiaofei Chen<sup>2,3</sup>

**Correspondence:** Yingjie Xia (xiayingjie@cdut.edu.cn)

Abstract. Correlating late coda waves from large earthquakes produces stable waveforms that approximate inter-station core phases. However, the properties of these coda waves often violate the strict assumptions underlying classical Green's function retrieval, raising doubts about the physical correspondence of the reconstructed arrivals to true inter-station phases and limiting their utility in seismic imaging. In this study, we present a perturbation analysis of core-phase interferometry and show that accurate travel-time information can be recovered under locally uniform wave incidence along the inter-station path. We introduce a dimensionless parameter — defined as the ratio of the seismic wave period to the inter-station travel time — which establishes a critical angular threshold. Our perturbation analysis reveals that the travel-time reconstruction accuracy scales with the cube of this threshold, allowing high-precision recovery of core phases, particularly those associated with small threshold values. Numerical simulations validate the theoretical predictions. By applying the proposed framework to real coda correlation data, we demonstrate that core phases can be reliably reconstructed using a sufficiently large number of global earthquakes — even without the traditionally assumed uniform source distribution. These results establish a rigorous theoretical foundation for extracting high-precision core-phase travel times from coda correlations, enhancing the reliability of seismological imaging of Earth's deep interior.

### 1 Introduction

Over the past two decades, the use of ambient ground motions for imaging subsurface structures has advanced significantly. This progress is largely driven by the discovery that cross-correlating records between two stations yields waveforms with dispersion characteristics resembling those of surface waves propagating between them (Campillo & Paul, 2003; Snieder, 2004). Commonly used ambient seismic sources include microseisms (1–50 period) and the Earth's hum(50–300 s), both generated by interactions between ocean waves and the solid Earth (Hasselmann, 1963; Ardhuin et al., 2015), as well as high-frequency anthropogenic noise (periods < 1 s). Dispersion measurements derived from ambient noise correlations have been extensively employed to probe Earth's interior structure (Shapiro et al., 2005; Sabra et al., 2005; Yao et al., 2006; Yang et al., 2007; Nishida et al., 2009; Wang et al., 2019).

<sup>&</sup>lt;sup>1</sup>Key Laboratory of Earth Exploration and Information Technology of Ministry of Education, Chengdu University of Technology, Chengdu, Sichuan, China, 610059

<sup>&</sup>lt;sup>2</sup>Shenzhen Key Laboratory of Deep Offshore Oil and Gas Exploration Technology, Southern University of Science and Technology, Shenzhen, Guangdong, China, 518055

<sup>&</sup>lt;sup>3</sup>Department of Earth and Space Sciences, Southern University of Science and Technology, Shenzhen, China, 518055

Beyond ambient noise, late earthquake coda waves contain substantial body wave energy that has traversed deep Earth discontinuities. Consequently, coda correlations are enriched with core-sensitive phases that generally preserve accurate slowness information (Lin et al., 2013; Nishida, 2013; Boué et al., 2014; Wu et al., 2018). Theoretically, the cross-correlation function (CCF) of noise records converges to the seismic Green's function under ideal conditions — such as a uniform distribution of noise sources enclosing the stations (Wapenaar, 2004) or equipartitioned wavefield energy (Lobkis & Weaver, 2001). However, coda waves often violate these ideal conditions, resulting in anomalously high amplitudes in reconstructed phases compared to earthquake data (Lin et al., 2013; Boué et al., 2014), as well as persistent features occurring at travel-time differences between conventional phases (Boué et al., 2014; Pham et al., 2018; Kennett & Pham, 2018a, b).

Deviations from ideal conditions are known to introduce nonphysical phases (Snieder et al., 2008) and also travel-time biases in surface wave reconstruction (Weaver et al., 2009; Tsai, 2009; Froment et al., 2010). Yet for coda-based core phases, a critical assessment of travel-time deviations between reconstructed and true inter-station arrivals remains lacking. Such an evaluation is essential to establish the reliability of extracted travel times for inferring deep Earth structural anomalies.

This study aims to evaluate the travel-time accuracy of core phases extracted from coda wave correlations. Previous theoretical studies on noise correlations often rely on asymptotic techniques — such as the stationary phase method — to establish approximate relationships, which limits the rigorous assessment of travel-time reconstruction accuracy. To address this, we introduce a perturbation-based approach to quantify potential travel-time deviations. A key aspect of this method is the decomposition of the problem into "solvable" and "perturbation" components. We begin with a bounded homogeneous model representing the solvable part, which is then perturbed to evaluate travel-time reconstruction accuracy. The proposed framework is validated through numerical simulations and demonstrated with real coda correlation data.

## 2 Theory

35

#### 2.1 The reference model

We consider a homogeneous medium bounded by two discontinuous surfaces to simulate wave reflections between the Earth's surface and the core-mantle boundary. The layer thickness is denoted by h. For simplicity, P-S wave conversion at the discontinuities is initially neglected, and the wave speed — for both P and S waves — is represented by a constant c. Two seismic stations are positioned at  $x_a = (0,0,0)$  and  $x_b = (R,0,0)$ , with all excitation sources placed within the surface layer (Fig. 1). For a wave originating from a source at position x = (x,y,0) and undergoing m reflections from the lower boundary before reaching station  $x_a$ , the ray path length is given by:

50 
$$r(x,m) = \sqrt{x^2 + y^2 + 4m^2h^2}$$
. (1)

Similarly, the path length for a wave arriving at  $x_b$  after experiencing n reflections is

$$r'(x,n) = \sqrt{(x-R)^2 + y^2 + 4n^2h^2} \ . \tag{2}$$

For a wave traveling directly between the two stations after undergoing p reflections, the path length is:

$$L(p) = \sqrt{R^2 + 4p^2h^2} \ . \tag{3}$$

The spectral representation of reflected waves recorded at either  $x_a$  or  $x_b$ , excited by a source located at x, can be expressed using a generalized ray formulation as:

$$u_{i}(\boldsymbol{x},\omega) = \sum_{m} A_{i}(\boldsymbol{x},m,\omega) e^{i\omega r(\boldsymbol{x},m)/c}$$

$$u'_{j}(\boldsymbol{x};\omega) = \sum_{n} A'_{j}(\boldsymbol{x},n,\omega) e^{i\omega r'(\boldsymbol{x},n)/c} .$$
(4)

In these equations,  $\omega$  represents the angular frequency, and i denotes the imaginary unit. The subscript i and j correspond to the three components of the displacement vector, respectively. The functions  $A_i(\boldsymbol{x}, m, \omega)$  and  $A'_j(\boldsymbol{x}, n, \omega)$  represent the amplitude of reflected waves. In this study, we restrict our analysis to incident angles below the critical angle; consequently, neither  $A_i(\boldsymbol{x}, m, \omega)$  nor  $A'_j(\boldsymbol{x}, n, \omega)$  incorporates a phase shift upon reflection and both remain real-valued.

We assume that the reflected wavefields excited by different sources are uncorrelated. Under this assumption, the total CCF, summed over all sources, can be expressed as:

$$C_{ij}(\omega) = \sum_{s} u_i^*(\boldsymbol{x}, \omega) u_j'(\boldsymbol{x}, \omega)$$
  
=  $\sum_{s} \sum_{m} \sum_{n} A_i(\boldsymbol{x}, m, \omega) A_i'(\boldsymbol{x}, n, \omega) e^{i\omega\psi(\boldsymbol{x}, m, n)}$ , (5)

where the summation over s corresponds to the contribution from all individual sources. The travel-time difference between the two ray paths is defined as:

$$\psi(\mathbf{x}, m, n) = \frac{r'(\mathbf{x}, n) - r(\mathbf{x}, m)}{c}$$

$$= \frac{1}{c} \left[ \sqrt{(x - R)^2 + y^2 + 4n^2h^2} - \sqrt{x^2 + y^2 + 4m^2h^2} \right].$$
(6)

## 2.2 The analytical solution of the CCF

The travel-time difference function in eq. (6) corresponds to the difference in travel times for wave propagation in a homogeneous medium. In this representation, the source is located at (x,y,2mh), and the two stations are positioned at (0,0,0) and (R,0,2ph), where p=m-n. Based on this equivalence, we evaluate the CCF in eq. (5) within this simplified homogeneous setting (Fig. 2).

Since the cases p < 0 and p > 0 are complex conjugate in the computation, we consider only p > 0 for simplicity. To facilitate the analysis, we apply a coordinate transformation by rotating the system about the y-axis so that the z-axis passes through the imaginary station at (R,0,2ph). In this rotated frame, we introduce spherical coordinates  $(r,\theta,\phi)$ . Note that the z-axis aligns with the reflected inter-station ray path originating from the station at (0,0,0); thus, the polar angle  $\theta$  represents the angular deviation of the incident wave from this ray path. Within this coordinate system, the travel-time difference function takes the

Figure 1. Definition of geometrical variables for wave propagation in the homogeneous medium bounded by two discontinuous layers.

form:

85

$$\psi(r,\theta,\phi,p) = \frac{1}{c}\sqrt{(r\sin\theta\cos\phi)^2 + (r\sin\theta\sin\phi)^2 + [r\cos\theta - L(p)]^2} - \frac{r}{c}$$

$$= \frac{r}{c}\sqrt{1 - \frac{2L(p)\cos\theta}{r} + \frac{L(p)^2}{r^2}} - \frac{r}{c}$$

$$= -\frac{L(p)}{c}\cos\theta \qquad (7)$$

80 The final approximation holds under the condition r >> L(p), which corresponds to m >> p.

In the CCF eq. (5), the variable pair (x,m) can be mapped to the spherical coordinates  $(r,\theta,\phi)$ , and similarly, (x,n) corresponds to  $(r,\theta,\phi,p)$ . To proceed, we introduce a continuous function  $\eta(r,\theta,\phi)$  to represent the density of the source distribution, which encapsulates the discrete contributions governed by the indices s and m. This allows the double summation over s and m to be approximated by a volume integral. Accordingly, the CCF can be rewritten as:

$$C_{ij}(\omega) = \sum_{s} \sum_{m} \sum_{p} A_{i}(r, \theta, \phi, \omega) A'_{j}(r, \theta, \phi, p, \omega) e^{i\omega\psi(r, \theta, \phi)}$$

$$= \sum_{p} \int_{0}^{\pi/2} \int_{0}^{2\pi} S_{ij}(\theta, \phi, p, \omega) e^{-ikL(p)\cos\theta} \sin\theta d\theta d\phi , \qquad (8)$$

where  $k = \omega/c$  is the wavenumber, and

$$S_{ij}(\theta,\phi,p,\omega) = \int_{r_1}^{r_2} \eta(r,\theta,\phi) A_i(r,\theta,\phi,\omega) A'_j(r,\theta,\phi,p,\omega) r^2 dr . \tag{9}$$

Figure 2. Definition of geometric variables for the case p = 1. The dashed circle and star denote the station and source mapped from  $x_b = (R, 0, 0)$  and (x, y, 0), respectively.

In the function  $S_{ij}(\theta,\phi,p,\omega)$ , we assume that the wave amplitudes at the two stations maintain a constant proportionality when excited by different sources. Under this assumption,  $S_{ij}(\theta,\phi,p,\omega)$  represents the wave energy density within each solid angle. We further assume that this energy density is uniform across all solid angles and denote it as  $\bar{S}_{ij}(p,\omega)$ . Consequently, the CCF can be expressed as:

$$C_{ij}(\omega) = \Sigma_p 2\pi \bar{S}_{ij}(p,\omega) \frac{e^{-ikL(p)\cos\theta}}{ikL(p)} \Big|_0^{\theta_0}$$

$$= 2\pi c \Sigma_p \bar{S}_{ij}(p,\omega) \left\{ \left[ \frac{e^{ikL(p)}}{i\omega L(p)} \right]^* - \left[ \frac{e^{ikL(p)\cos\theta_0}}{i\omega L(p)} \right]^* \right\}.$$
(10)

Here,  $\theta_0$  represents upper boundary of the polar angle. We assume it is azimuthally symmetric (independent of  $\phi$ ). In the time domain, the first term of this equation corresponds to arrivals at the travel times of the reflected waves between the two stations, where the factor  $1/i\omega$  corresponds to a time-domain integration operator. The second term corresponds to spurious waves arising due to a uniform truncation of the polar angle at different azimuthal angles. When p 

125

This yields a threshold angle expression:

105 
$$\theta_0 = 2\arcsin\sqrt{\frac{\lambda}{2L(p)}}$$
 i.e.,  $\theta_0 = 2\arcsin\sqrt{\frac{T}{2t(p)}}$ . (12)

Here, T denotes the period of the reflected wave and t(p) represents the travel time of the inter-station wave that has undergone p reflections. This result defines the minimum angular range of wave incidence required to maintain local wavefield uniformity, thereby ensuring accurate wave reconstruction.

# 2.3 Perturbation analysis for realistic coda correlations

The derivation above assumes a cosine distribution for the travel time difference function, which depends on wave propagation taking place within a bounded homogeneous medium. In practice, this condition is not satisfied. Furthermore, late earthquake coda correlations also involve P-to-S wave conversions. Under these circumstances, the actual travel time difference function deviates from the cosine form. To accommodate such deviations, we express the perturbed travel time differences as:

$$\psi(r,\theta,\phi,p) = -t(p)\cos\theta + \delta(\theta,p) , \qquad (13)$$

where t(p) denotes the travel time along curved ray paths between the two stations after p reflections,  $\theta$  is the polar angle between the incident wave direction and the z-axis (where the z-axis is aligned with the direction of the inter-station reflected wave at the station), and  $\delta(\theta,p)$  captures deviations from the idealized cosine distribution. Our analysis assumes azimuthal symmetry (i.e., independent of  $\phi$ ) and ignores the dependence on travel distance r for simplicity. The formalism can be readily extended to incorporate such dependencies by performing the analysis over discrete values of  $\phi$  and r.

Since  $\theta = 0$  corresponds to the inter-station ray path, the travel time difference at this angle attains its extreme value. We impose:

$$\delta(0,p) = 0$$
 and  $\delta^{(1)}(0,p) = 0$ , (14)

where the superscript (n) denotes the n-th derivative with respect to  $\theta$ . Expanding  $\delta(\theta, p)$  in a Taylor series around  $\theta = 0$ , the travel time difference becomes:

$$\psi(r,\theta,\phi,p) = -t(p)\cos\theta + \frac{1}{2}\delta^{(2)}(0,p)\theta^2 + \frac{1}{6}\delta^{(3)}(0,p)\theta^3 + \frac{1}{24}\delta^{(4)}(0,p)\theta^4 + \cdots$$

$$= \delta^{(2)}(0,p) - [\delta^{(2)}(0,p) + t(p)]\cos\theta + \frac{1}{6}\delta^{(3)}(0,p)\theta^3 + \frac{1}{24}[\delta^{(2)}(0,p) + \delta^{(4)}(0,p)]\theta^4 + \cdots,$$
(15)

where the second line follows from substituting the Taylor expansion of  $\cos \theta$ .

Truncating the series at the  $\theta^3$  term and substitute into eq. (8) (assuming that the polar angle is truncated at  $\theta_0$ , beyond which wave construction is not affected) yields:

$$C_{ij}(\omega) = \sum_{p} \int_{0}^{\theta_{0}} \int_{0}^{2\pi} S_{ij}(p,\omega) e^{i\omega\{\delta^{(2)}(0,p) - [\delta^{(2)}(0,p) + t(p)]\cos\theta\}} \sin\theta d\theta d\phi$$

$$= \sum_{p} \frac{2\pi S_{ij}(p,\omega)}{i\omega[\delta^{(2)}(0,p) + t(p)]} e^{i\omega\delta^{(2)}(0,p)} \cdot e^{-i\omega[\delta^{(2)}(0,p) + t(p)]\cos\theta} \Big|_{0}^{\theta_{0}}$$

$$= \sum_{p} \frac{2\pi S_{ij}(p,\omega)}{\delta^{(2)}(r,0,p) + t(p)} \left\{ \frac{e^{i\omega[\delta^{(2)}(0,p)(1 - \cos\theta_{0}) - t(p)\cos\theta_{0}]}}{i\omega} + \left[\frac{e^{i\omega t(p)}}{i\omega}\right]^{*} \right\}.$$
(16)

This result shows that the correlation still yields waves at the travel times of the reflected waves when truncating the series of the deviation function  $\delta(\theta, p)$  at  $\theta^3$ . We adopt, as before, our non-interference criterion for the phase difference. This leads to the condition:

$$\omega t(p) + \omega [\delta^{(2)}(0, p)(1 - \cos \theta_0) - t(p)\cos \theta_0] \ge 2\pi . \tag{17}$$

We obtain a threshold angle expression:

135 
$$\theta_0 = 2\arcsin\sqrt{\frac{T}{2[t(p) + \delta^{(2)}(0, p)]}}$$
, (18)

The second derivative  $\delta^{(2)}(r,0,p)$  affects the polar angle range over which the wavefield needs to be locally uniform. If we neglect its impact in this equality, the expression reduces to equality (12).

Equation (16) demonstrates that time errors due to deviations from the cosine distribution arise from higher-order terms ( $\theta^3$  and beyond) in the Taylor series expansion of the travel time difference function  $\psi(r,\theta,\phi,p)$ . We estimate the time error as:

140 
$$\Delta t \approx \frac{1}{6} \delta^{(3)}(0, p) \theta_0^3 + \frac{1}{24} [\delta^{(2)}(0, p) + \delta^{(4)}(0, p)] \theta_0^4 + \cdots,$$
 (19)

which scales proportionally to  $\theta_0^3$ . For the reconstruction of core phases, such as the ScS wave with a period of 50 s and a travel time of 1000 s, the threshold angle is:

$$\theta_0 = 2\arcsin\sqrt{\frac{50}{2 \times 1000}} = 18^\circ \ . \tag{20}$$

Converting this angle to radians and substituting into eq. (19) yields a time variation on the order of:

145 
$$\Delta t \approx \frac{\delta^{(3)}(0,p)}{220}$$
. (21)

Consequently, if  $\delta(\theta, p)$  is smooth near  $\theta = 0$  (i.e., its derivatives are small), the resulting time deviation is negligible. This result demonstrates that the core phases can be accurately reconstructed via coda correlation under the assumption of local wavefield uniformity.

Figure 3. The computations of the CCF under three truncation angles.

## 3 Numerical simulations

160

We perform a numerical computation to investigate the impact of wave correlation under localized incidence uniformity. In the computation, we set the travel time of the reflected wave to 1000 s and wave correlation in the period range of 20–50 s. Then, we obtain the threshold angle  $\theta_0 = 18^{\circ}$  as in eq. (20), corresponding to a  $2\pi$  phase shift relative to the reflected wave. We also compute truncation angles for  $\pi$  and  $4\pi$  phase shifts, obtaining  $\theta_0 = 13^{\circ}$  and  $\theta_0 = 26^{\circ}$ , respectively. These truncation angles are used as upper bounds in the integral of eq. (10). For comparison, we compute the accurate arrival of the reflected wave using an upper bound of  $180^{\circ}$ . The results show: Within a  $\pi$  phase shift, the reconstructed reflected wave and truncation-induced spurious wave interfere, causing phase deviation in the reconstructed reflected wave. Within a  $2\pi$  phase shift, the two waves align, allowing accurate recovery of the reflected wave's travel time. Within a  $4\pi$  phase shift, the waves diverge, and the reconstructed reflected wave remains unaffected by integration truncation (Fig. 3). This supports the rationality of our non-interference criterion that uses a  $2\pi$  phase shift to determine the threshold angle.

Given the small threshold angle in our simulation, we further examine correlation under a non-cosine travel-time difference distribution by setting the disturbance term in eq. (13) as

$$\delta(\theta, p) = -50\theta \sin^2(4\theta). \tag{22}$$

**Figure 4.** The computation of the CCF under a non-cosine distribution of travel time differences. (a) The travel time difference function; (b) The simulated CCF.

This function and its first derivative satisfy the constraints in eq. (14). Integrating over  $\theta$  from 0 to  $\pi$  and comparing the reconstructed wave with that from a cosine travel-time difference distribution, we find nearly identical phase information (Fig. 4). This demonstrates high reconstruction accuracy for travel times even under non-cosine travel-time difference distributions.

# 4 The real-data test

165

In the analysis of late earthquake coda correlations, when the plane containing the earthquake and the station is oriented at an angle  $\phi$  relative to the two-station plane, the following geometric relationship applies:

$$\sin \phi = \frac{\sin \theta}{\sin i},\tag{23}$$

where i represents the incident angle of the wave at the station. Core phases in late coda correlations are characterized by steep incident angles. As a result, even a small threshold angle  $\theta_0$  corresponds to a wide range of azimuthal deviations from the great-circle plane (see Fig. 5a). For instance, for an ScS wave with an incident angle  $i=20.0^\circ$  and a threshold angle  $\theta_0=18^\circ$ , equation (23) gives  $\phi\approx65^\circ$ . In cases where  $\theta>i$ , even coda waves propagating in a plane perpendicular to the two-station plane can contribute to the reconstruction of core phases (Fig. 5b). This result demonstrates the importance of incorporating earthquakes from all azimuths to ensure accurate reconstruction of core phases.

We analyze codas from 205 large earthquakes ( $\geq$  M6.8) recorded by the US network between 2010 and 2020 (Fig. 6). The correlation process is as described in the work of Bensen et al. (2007), which includes: filter the coda data to 15–50 seconds, suppress the records by temporal normalization and spectral whitening, and independently compute CCFs for each earthquake.

**Figure 5.** Relationship between the threshold angle and azimuth: (a) when the threshold angle is less than the wave incident angle, and (b) when the threshold angle exceeds it. In both cases, the gray line denotes the extended trajectory of the inter-station ray path, whereas the shaded region illustrates the area encompassed within the threshold angle.

The computed CCFs are then categorized into bins according to the inter-station distance, with each bin encompassing an interval of 1.0 degree. Finally, the CCFs are stacked together within each bin.

In the stacked correlograms, prominent deep phases such as PcP, ScS, and  $PKIKP^2$  waves are clearly distinguishable (Fig. 7). To assess the travel time accuracy of reconstructed core phases, we stack correlograms for earthquakes occurring in two distinct periods: 2010–2015 and 2015–2020, which exhibit varying earthquake positions (see Fig. 6a). Despite these differences, both stacked correlograms show nearly identical emergence times for core phases, indicating stable convergence of the reconstructed core phases under different source distribution (Fig. 8).

To investigate the azimuthal dependence of the stacked deep reflections, we compute the deviation angle  $\phi$  between the earthquake-station plane and the plane defined by the station pair. Correlograms are then stacked within selected ranges of  $\phi$ . Significant disparities in the emergence times of the ScS waves are observed when comparing stacks for a narrow azimuthal range ( $\phi 

200

205

Figure 6. (a) The distribution of used large earthquakes (circles) and stations (triangles), and (b) the zoom-in station distribution.

Figure 7. The stacked correlograms. Some prominent phases are labeled.

time, quantifies the extent of this localized uniformity. Equation (19) further shows that travel time reconstruction accuracy scales with the cube of this threshold angle. Since core phases are characterized by inherently small threshold angles, high travel time extraction accuracy is achieved under localized uniformity conditions.

Seismological observations indicate that incident energy in late coda waves is predominantly concentrated near the plane between the earthquake and the station (Sens-Schönfelder et al., 2015). A limited number of earthquakes located near this plane can satisfy the localized uniformity condition around the inter-station ray path in the great-circle direction. Consequently, our approach relaxes the stringent requirement for full source uniformity along the great circle — necessary in traditional Green's function retrieval — making it more applicable to practical coda correlation studies.

Figure 8. Comparison of the stacked correlograms within the two distinct time periods: 2010-2015 and 2015-2020.

**Figure 9.** Comparison of the stacked correlograms using earthquakes with  $\phi$  in  $(0, 10^{\circ})$  and using all earthquakes.

210

Figure 10. The convergence of ScS and  $PKIKP^2$ -like waves as the deviation angle ranges increase. The bin size for the inter-station distance is  $10^{\circ}$ . The time window for phase calibration is indicated within the dashed box, and the waveform is normalized based on the wave amplitude within this window.

Equation (23) shows that, due to the steep incidence angles of core phases, the small threshold angle for core phases corresponds to a considerable range of azimuthal deviations from the great-circle plane. Therefore, incorporating earthquakes from all azimuths is recommended to ensure accurate reconstruction. In real-data tests of  $PKIKP^2$  phase reconstruction, waveforms derived from earthquakes within a narrow angular range aligned closely with those obtained using all available events. This consistency is likely due to lateral scattering of late coda waves during propagation, which effectively samples the required extremely small threshold angle range. These results suggest that  $PKIKP^2$  waves can be reliably reconstructed using earthquakes located near the great-circle plane connecting the station pair.

To ensure localized wavefield uniformity in our real-data test, we employed a large global dataset of earthquakes. As there is no direct method to confirm whether reconstructed core phases correspond exactly to true inter-station arrivals, we divided the earthquake records into two distinct time periods and compared the phase alignment between independently reconstructed waveforms. Close agreement between the two subsets indicates successful recovery of the true inter-station core phase. This convergence-based criterion provides a statistical validation approach to assessing the accuracy of reconstructed core phases.

Equation (19) quantifies the dependence of inter-station body wave reconstruction accuracy on the wave period-to-travel time ratio. It reveals that travel time deviations arise even under uniform illumination — a finding consistent with surface wave

dispersion studies in inhomogeneous media (Tsai, 2009). This relationship offers a criterion for assessing travel time deviations in reconstructed waves, which is especially valuable when the ratio of wave period to the inter-station travel time ratio is small.

According to eq. (10), the correlation process can generate spurious waves due to uniform truncation of  $\theta$  across azimuths. Since such truncation is rarely achieved in practice, spurious contributions are summed over various azimuths. When the truncation boundary extends beyond the stationary phase zone, destructive interference generally suppresses these spurious waves. However, near  $\theta = \pi/2$ , an inflection point effect inhibits cancellation (Xia et al., 2024) and may introduce spurious arrivals near zero time on the CCF if wave incidence intensity is asymmetric with respect to the station pair.

Finally, this study employs a simplified model to conduct a perturbation analysis of travel-time reconstruction accuracy. For more realistic scenarios — such as multilayered media with smoothly varying wave speeds — the generalized ray method offers a suitable framework to simulate wave conversion, thereby facilitating the extraction of both inter-station body waves and the persistent unconventional waves observed in practical coda correlations. However, such models do not readily permit a decomposition comparable to that of the perturbation approach, which is why this more complex scenario is not addressed in the current study.

#### 6 Conclusions

230

235

240

This study presents a perturbation analysis to evaluate the accuracy of travel-time reconstruction for core phases derived from late earthquake coda correlations under conditions of locally uniform wave incidence along the core-phase propagation path. We introduce a dimensionless parameter, defined as the ratio of seismic wave period to inter-station travel time, to quantify the critical angular threshold for effective reconstruction. Perturbation analysis reveals that the travel time accuracy scales with the cube of this threshold, indicating that localized uniform incidence ensures high-precision reconstruction of core phases, which are inherently characterized by low threshold values. Numerical simulations and empirical coda correlation tests sustain our theoretical findings. Our results demonstrate that accurate travel times of inter-station core phases can be reliably extracted using late coda waves from a sufficiently large number of earthquakes distributed across all azimuths. This approach provides a practical and robust foundation for coda correlation studies, enhancing confidence in using reconstructed core phases as true empirical arrivals for interferometric imaging of Earth's deep interior.

Data availability. Seismic data are from the IRIS Data Management Center: https://ds.iris.edu/ds/nodes/dmc/data/types/waveform-data/.

Acknowledgements. This work was supported by Shenzhen Offshore Oil and Gas Exploration Technology (Grant ZDSYS20190902093007855), the National Natural Science Foundation of China (Grants 41790465, 42241141 and 42374072)

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
