# Peer review of "Perturbation Analysis of Travel-Time Accuracy for Core Phases Reconstructed from Seismic Interferometry"

_EGUsphere, 2025_

## Author Comment (AC1)

This manuscript tackles a pivotal issue in seismology: enhancing the accuracy of core-phase travel-time reconstruction through coda wave interferometry, a topic of paramount importance for advancing the imaging of Earth's deep interior. The authors employ a perturbation analysis to quantify travel-time deviations and establish a critical angular threshold, constituting a valuable theoretical contribution that addresses persistent uncertainties regarding the physical correspondence between reconstructed coda-derived phases and true inter-station core phases. However, several substantive issues pertaining to literature integration, theoretical rigor, completeness in data analysis, and contextualization with prior research must be resolved to bolster the manuscript's robustness and impact. Based on the following assessment, a major revision is recommended.

Thank you for the thorough review and positive assessment of our work. We appreciate your recognition of the study's core contribution and your constructive identification of areas for improvement. We agree that addressing the key issues you raised will strengthen the manuscript's robustness.

We have carefully reviewed all your specific comments. In the revised version, we will address each point. We sincerely thank you for your time and valuable feedback, which are essential for improving this work.

Major revisions:

1, Insufficient Quantitative Analysis of Data Volume, Distribution, and Deviation Angle Dependence on Reconstruction Stability:

A central conclusion of this study is that "a sufficiently large number of global earthquakes" enables reliable core-phase reconstruction. However, this claim lacks the necessary quantitative foundation and fails to systematically assess how the stability of travel-time retrieval depends on key data characteristics. Specific shortcomings include:

a) Undocumented Processing Parameter: The manuscript does not specify the coda time window (e.g., start and end times relative to the origin time) used for the correlation analysis. This critical parameter directly influences the extracted waveforms and must be reported to ensure reproducibility.

In the revised manuscript, we have clarified that for each large earthquake, the late coda segment used for correlation is defined as the window from 10,000 to 40,000 seconds following the origin time.

b) Unquantified Impact of Event Count: While 205 earthquakes were used, the study provides no analysis of how reconstruction stability degrades with smaller datasets. To objectively define data sufficiency, a sensitivity analysis is required. For instance: How does the standard deviation of the reconstructed travel times for key phases (e.g., ScS, $PKIKP^2$) evolve as the number of events used in the stack is progressively reduced from 205 to, for example, 150, 100, or 50 (e.g., via bootstrap resampling)? Does a minimum event count threshold exist, below which the stability deteriorates significantly or the proposed cubic scaling relationship between accuracy and the angular threshold breaks down?

In the revised manuscript, we examine the convergence of travel-time measurements as the number of earthquakes is progressively reduced. Bootstrap resampling is used to assess the associated travel-time deviations. Convergence serves as an indicator that the local uniformity condition is satisfied and, consequently, that the extracted travel times represent accurate estimates of true inter-station travel times.

[Figure]

The figure above illustrates the evolution of ScS travel time as the number of earthquakes included in the correlation (stacking bin 10°–11°) decreases. Significant deviations in the constructed ScS wave emerge when fewer than 20 earthquakes are used. However, uniformity is influenced not only by the earthquake count but also by the number of bin-stacked correlation traces (here, 67 traces). Therefore, in the revised text, we avoid designating a fixed threshold of 20 earthquakes as a guarantee for accurate travel-time construction.

Our theoretical analysis derives a cubic scaling relationship based on the assumption of uniform wave incidence. In cases where the earthquake count is insufficient and the bin-stacking procedure cannot achieve such uniformity, this scaling relationship may no longer hold.

c) Incomplete Investigation of Deviation Angle Dependence: While Figure 10 presents a valuable analysis of convergence with increasing maximum deviation angle ($\varphi$), the current approach of using cumulative ranges (e.g., 0-10°, 0-20°, etc.) limits its power to validate the theoretical framework. A more rigorous, binned analysis is needed. The data should be stacked and analyzed within discrete, non-overlapping ranges of the deviation angle $\varphi$ (e.g., 0-10°, 10-20°, 20-30°, etc.).

This binned comparison is critical for directly testing the theoretical prediction of whether travel times remain accurate and unbiased across all directions of wave incidence under the local uniformity assumption.

Such an analysis would reveal: 1) If travel-time biases exist in specific ranges of the deviation angle, and 2) The magnitude of any such biases as a function of $\varphi$.

A finding of consistent travel times across all discrete angular bins would strongly corroborate the theoretical model. Conversely, identifying systematic biases in certain angular ranges would provide invaluable insights into the limits of the local uniformity condition and guide future data selection.

Thank you for this constructive and detailed suggestion. In the revised manuscript, we have adopted your recommended method and now present the correlation functions stacked within **discrete, non-overlapping ranges of the deviation angle φ** .

As shown in the new figure, the results clearly demonstrate a strong dependence of ScS travel times on the deviation angle, while PKIKP2 travel times remain relatively stable.

We attribute this fundamental discrepancy directly to the **ratio between the incident wave angle and the critical angle ($\theta_0$),** as predicted by our theoretical model. For ScS, the incident angle is relatively large compared to its critical angle, corresponding to the scenario in Figure 5(a) where travel-time reconstruction is sensitive to the angular distribution of sources. For PKIKP2, the incident angle is relatively small compared to its larger critical angle, corresponding to the stable scenario in Figure 5(b).

Thus, this analysis not only confirms the empirical contrast but is also fully explained by the current theoretical model, providing a direct test of its predictive power regarding the limits of the local uniformity condition for different phases.

[Figure]

2, Missed Opportunity for Theoretical Validation through Comparative Analysis of ScS and PKIKP² Phases

The manuscript estimates a critical angle of 18° for the ScS phase and applies this framework in the real-data analysis. However, it does not fully leverage the contrasting behaviors of ScS and PKIKP² phases observed in Figure 10 to rigorously test and validate the underlying theory. Specifically, the ScS travel time shows a clear dependence on the deviation angle φ, while the PKIKP² travel time remains stable. This

striking discrepancy represents a critical opportunity to strengthen the study's conclusions, yet it remains largely unexplained.

To transform this observation into a powerful validation of the theoretical framework, the authors should:

a) Perform Phase-Specific Theoretical Calculations: The critical angle and the expected travel-time deviation are functions of the wave period (T) and the phase-specific travel time (t(p)). The manuscript must present the theoretically predicted critical angle and the scaling of travel-time accuracy specifically for the PKIKP² phase, rather than implicitly assuming the ScS-derived value applies.

In the revised manuscript, we have conducted theoretical calculations for the PKIKP² phase. This includes determining its critical angle $\theta_0$ and small incidence angle, which differ from those of ScS due to its distinct ray path and travel time. By presenting these results, we provide a direct physical explanation for the observed contrast in sensitivity to deviation angles between the ScS and PKIKP² phases.

b) Provide a Physically Consistent Explanation for the Contrast: The fundamental difference in the sensitivity of ScS and PKIKP² to the deviation angle $\varphi$ must be explained within the proposed theoretical framework. This discussion should explicitly link the distinct ray paths of

the two phases (e.g., ScS reflecting off the core-mantle boundary versus PKIKP² traversing the inner core) to the potential magnitude of the deviation function $\delta(\theta, p)$ and its higher-order derivatives in Eq. 19. For instance:

Does the more complex path of PKIKP² through the inner core lead to a different "smoothness" of $\delta(\theta, p)$ near $\theta=0$, resulting in smaller higher-order terms and thus greater robustness to a limited deviation angle range?

Conversely, does the ScS path make it more susceptible to structural heterogeneity near the core-mantle boundary, amplifying the higher-order derivatives and making its reconstruction more sensitive to the angular distribution of sources?

By quantitatively calculating the phase-specific theoretical parameters and then using them to explain the empirically observed difference in stability between ScS and PKIKP², the authors can demonstrate that their model not only predicts general behavior but also accurately captures the specific physics governing different core phases. This would significantly elevate the impact of the study by providing a unified and predictive theoretical explanation for the key observational result in Figure 10.

In the revised manuscript, we will expand our theoretical explanation along the lines you suggested. Currently, our analysis provides a qualitative explanation for the observed stability difference, attributing it to the ratio between the wave incidence angle and the phase-specific critical angle. This ratio determines the fraction of useful signal that falls within the stable angular range (Figure 5), with waves inside this range contributing to stable travel-time recovery.

Conceptually, if we treat this stable angular range as a stationary phase zone, our current analysis is based on the second-order approximation of the travel-time difference function. We have chosen not to extend the analysis to higher-order terms of $\delta(\theta, p)$, as doing so would introduce greater theoretical uncertainty without providing proportionally clearer insight.

3, Unaddressed Discrepancies in Figures 8 and 9: Both figures indicate that most core phases exhibit obvious waveform differences as the inter-station distance approaches 0°. The manuscript does not investigate the origin of this systematic pattern, which could stem from physical phenomena (e.g., near-field effects, 3D structural complexities) or methodological artifacts (e.g., inadequate azimuthal coverage at very short distances). Explaining this observation is vital for affirming the method's reliability across the entire distance range.

Thank you for this important observation regarding the systematic waveform differences at short inter-station distances in Figures 8 and 9.

In the revised manuscript, we will expand our discussion on this point. As noted in our response, the primary cause is indeed the **low signal-to-noise ratio (SNR) of the reconstructed core phases at very short distances**. We will elaborate on the underlying physical mechanism, as derived from our theoretical framework: the constructive interference necessary to build a coherent signal for a specific phase (e.g., ScS) depends critically on the path difference between the two stations. At near-zero distances, the wave propagation energy on such a path is weak, reducing the amplitude of the target phase in the cross-correlation stack and thus lowering its SNR.

By providing this clear explanation, we demonstrate that the observed pattern is a predictable consequence of the method.

4, Need for a Unifying Theoretical Discussion on I2* versus True Phase Reconstruction for PKIKP²:

The manuscript reports stable PKIKP² travel times across a wide range of deviation angles (Fig. 10). This finding appears to contradict a body of

prior work (e.g., Wang & Tkalčić, 2019, 2020; Costa de Lima et al., 2022) which argues that coda correlations typically retrieve an I2* wavefield—a modified Green's function whose travel times exhibit a dependence on the distribution of seismic sources (e.g., varying with deviation angle). The authors have a critical opportunity to use their theoretical framework to explain and reconcile these differing observations, thereby making a seminal contribution to the debate on what is physically extracted from coda correlations.

To achieve this, the authors must:

a) Explicitly Discuss the Discrepancy within Their Theoretical Context: The discussion should directly engage with the findings of the aforementioned I2* studies. The core argument should posit that the critical difference lies in whether the condition of "local uniform wave incidence" (quantified by the critical angle $\theta_0$) stem from datasets or phases where this local uniformity condition was not satisfied. In contrast, the current study, potentially by leveraging a massive global dataset for $PKIKP^2$, may have met this condition, thus successfully retrieving the true inter-station travel time.

Thank you for raising this point. We agree that reconciling previous I2* results with our findings is critical for a complete theoretical narrative.

In the revised discussion, we will directly engage with the cited studies on I2* by framing the core discrepancy within our theoretical context. We will argue that the key difference lies in whether the condition of **"local uniform wave incidence"**—quantified by the phase-specific critical angle $\theta_0$—is satisfied. Specifically, we will propose that previous observations of a biased I2* phase likely occurred in datasets or phases where this local uniformity condition was not met, while the stable $PKIKP^2$ result in our study may be attributed to a sufficiently large and well-distributed dataset that satisfied this condition.

This argument will be further supported by our new analysis on travel-time convergence with earthquake count, as suggested in Comment 1(a). The observed convergence strengthens the reasoning that a massive global dataset can achieve the necessary illumination to retrieve the true inter-station travel time, thereby reconciling the two seemingly contradictory bodies of work.

b) Provide a Physical Mechanism for $PKIKP^2$ Stability Based on the Perturbation Analysis: The authors must use their theoretical framework to explain why the $PKIKP^2$ phase in their study is robust. The key lies in Eq. (19): the travel-time error scales with $\theta_0^3$ and the higher-order derivatives of the deviation function $\delta(\theta, p)$.

The authors should argue that the specific ray path of PKIKP² (traversing the inner core) results in a "smoother" $\delta(\theta,p)$ near $\theta=0$ (i.e., very small higher-order derivatives). Combined with its specific period-to-travel-time ratio yielding a small $\theta_0$, this makes the phase inherently robust to variations in the deviation angle $\varphi$ once a basic illumination threshold is crossed.

This provides a physical mechanism for why their method, under the right conditions, avoids the deviation-angle-dependent biases characteristic of I2* retrieval.

Thank you for this insightful suggestion. In the revised manuscript, we have further elaborated on the physical mechanism for PKIKP² stability within our theoretical framework.

Our explanation centers on the **fraction of useful signal that falls within the stable angular range**, as defined by the phase-specific critical angle $\theta_0$. Signals within this range constructively interfere to yield stable travel-time measurements. For PKIKP², due to its small effective incidence angle relative to $\theta_0$, a significant portion of the coda energy—even from earthquakes at larger deviation angles—contributes to this stable reconstruction (Figure 5b). This directly accounts for the observed robustness.

In our perturbation analysis, the confinement of the stable angular range in Eq. (18) already incorporates the second-order behavior of $\delta(\theta,p)$. Extending the explanation to explicitly discuss higher-order terms of $\delta(\theta,p)$—while theoretically interesting—would introduce significant uncertainty in our current framework, as $\delta(\theta,p)$ is difficult to characterize analytically after multiple coda reflections. Therefore, we maintain that the stability of $PKIKP^2$ is most clearly and reliably explained by the angular signal fraction mechanism derived from our second-order analysis.

c) Propose a Generalized Criteria for True Travel-Time Extraction: The manuscript should synthesize these points into a clear proposition: The transition from retrieving a biased I2 to the true $PKIKP^2$ travel time occurs when the angular range of incident waves meets or exceeds the phase-specific critical angle $\theta_0$ and the structural setting leads to a sufficiently smooth deviation function. This would powerfully contextualize their results, suggesting that the previous I2* observations and their own stable result are not fundamental contradictions but are explained by the degree to which the conditions of their unified theory are met.

Thank you for this insightful suggestion. In the revised discussion section, we have incorporated a concise proposition as you recommended, explicitly stating that the transition from a biased I2 measurement to a true PKIKP² travel time requires: 1) an angular range of incident waves meeting or exceeding the phase-specific critical angle $\theta_0$, and 2) a structural setting that yields a sufficiently smooth travel-time deviation function. We further clarify how this framework reconciles previous I2* observations with our stable results, treating them as outcomes determined by the degree to which these unified conditions are met.

Minor comments:

1, The Introduction's discussion of the research gap concerning travel-time deviations for coda-based core phases requires sharper focus and better integration with key literature. Notable studies on core-phase extraction (e.g., Wang & Tkalčić, 2019, JGR-Solid Earth; Poli et al., 2017, Earth and Planetary Science Letters; Phạm & Tkalčić, 2022, Nature Communications) should be cited to delineate the knowledge gap and underscore the novelty of this work.

Cited

2, To strengthen the practical motivation, the Introduction should explicitly mention applications of coda-based core phases in imaging

Earth's interior. Citing relevant studies (e.g., Wang, Song, & Xia, 2014, Nature Geoscience; Tkalčić & Phạm, 2018, Science; Wang & Tkalčić, 2022, Nature Astronomy) would highlight the significance of accurate travel-time reconstruction.

We have cited these papers.

3, Several conclusions would benefit from citations to post-2020 research to enhance timeliness. For instance, referencing recent comparative studies on core-phase travel-time accuracy (e.g., Costa de Lima et al., 2022, JGR-Solid Earth) would help contextualize the findings within the latest advancements.

Thank you for the suggestion. We update the relevant sections by citing recent comparative studies, to enhance the manuscript's context.

4, The terminology for the angle of wave arrival at a station is inconsistent, alternating between "incident angle" and "incidence angle." The standard seismological term "incidence angle" should be used consistently throughout the manuscript.

Revised

5, The manuscript uses "azimuth" to describe the orientation of the earthquake-station plane relative to the inter-station plane. This usage is

not explicitly distinguished from the standard seismological definition of azimuth (the angle from true north to the source-station great-circle path). To prevent confusion, the authors should clearly define their custom azimuth (or use "deviation angle") and clarify its relationship to the conventional term.

In the revision, we use deviation angle to replace "azimuth". Thanks for this recommendation.

6, "microseisms (1–50 period)" should be "microseisms (periods 1–50 s)"

Revised

7,"205 large earthquakes (≥M 6.8)" should be "205 large earthquakes (M ≥ 6.8)"

Revised